# Distributed Training of Large Graph Neural Networks with Variable Communication Rates

## Abstract

Training Graph Neural Networks (GNNs) on large graphs presents unique challenges due to the large memory and computing requirements. Distributed GNN training, where the graph is partitioned across multiple machines, is a common approach to training GNNs on large graphs. However, as the graph cannot generally be decomposed into small non-interacting components, data communication between the training machines quickly limits training speeds. Compressing the communicated node activations by a fixed amount improves the training speeds, but lowers the accuracy of the trained GNN. In this paper, we introduce a variable compression scheme for reducing the communication volume in distributed GNN training without compromising the accuracy of the learned model. Based on our theoretical analysis, we derive a variable compression method that converges to a solution that is equivalent to the full communication case. Our empirical results show that our method attains a comparable performance to the one obtained with full communication and that for any communication budget, we outperform full communication at any fixed compression ratio.

## 1 Introduction

Graph Neural Networks (GNNs) are a neural network architecture tailored for graph-structured data (Zhou et al., 2020; Wu et al., 2020; Bronstein et al., 2017). GNNs are multi-layered networks, where each layer is composed of a (graph) convolution and a point-wise non-linearity (Gama et al., 2018). GNNs have shown state-of-the-art performance in robotics (Gama et al., 2020a; Tzes et al., 2023), weather prediction (Lam et al., 2022), protein interactions (Jumper et al., 2021) and physical system interactions (Fortunato et al., 2022), to name a few. The success of GNNs can be attributed to some of their theoretical properties such as being permutation-invariant (Keriven and Peyré, 2019; Satorras et al., 2021), stable to perturbations of the graph (Gama et al., 2020b), transferable across graphs of different sizes (Ruiz et al., 2020), and their expressive power (Xu et al., 2018; Bouritsas et al., 2022; Kanatsoulis and Ribeiro, 2022; Chen et al., 2019).

In a GNN, the data is propagated through the graph via graph convolutions, which aggregate information across neighborhoods. In large-scale graphs, the data diffusion over the graph is costly in terms of computing and memory requirements. To overcome this limitation, several solutions were proposed. Some works have focused on the transferability properties of GNNs, i.e. training a GNN on a small graph and deploying it on a large-scale graph (Ruiz et al., 2020; Maskey et al., 2023; 2022). Other works have focused on training on a sequence of growing graphs (Cerviño et al., 2023; Cerviño et al., 2023). Though useful, these solutions either assume that an accuracy degradation is admissible (i.e. transferability bounds), or that all the graph data is readily accessible within the same training machine. These assumptions may not hold in practice, as we might need to recover the full centralized performance without having the data in a centralized manner.

Real-world large-scale graph data typically cannot fit within the memory of a single machine or accelerator, which forces GNNs to be learned in a distributed manner (Cai et al., 2021; Wang et al., 2021; Zheng et al., 2020; Wang et al., 2022). To do this efficiently, several solutions have been proposed. There are *data-parallel* approaches that distribute the data across different machines where model parameters are updated with local data and then aggregated via a parameter server. Another solution is *federated learning*(FL), where the situation is

even more complex as data is naturally distributed across nodes and cannot be shared to a central location due to privacy or communication constraints(Bonawitz et al., 2019; Li et al., 2020a;b). Compared to data parallel approaches FL suffers from data heterogeneity challenges as we cannot control the distribution of data across nodes to be identically distributed(Shen et al., 2022). The GNN-FL adds additional complexity as the graph itself (input part of the model) is split across different machines. The GNN FL counterpart has proven to be successful when the graph can be split into different machines(He et al., 2021; Mei et al., 2019). However, training locally while assuming no interaction between datasets is not always a reasonable assumption for graph data.

This work is drawn by two observations of GNNs. First, large graph datasets cannot be split into non-interacting pieces across a set of machines. Therefore, training GNNs distributively requires interaction between agents in the computation of the gradient updates. Second, the amount of communicated data affects the performance of the trained model; the more we communicate the more accurate the learned model will be. In this paper, we posit that the compression rate in the communication between agents should vary between the different stages of the GNN training. Intuitively, at the early stages of training, the communication can be less reliable, but as training progresses, and we are required to estimate the gradient more precisely, the quality of the communicated data should improve.

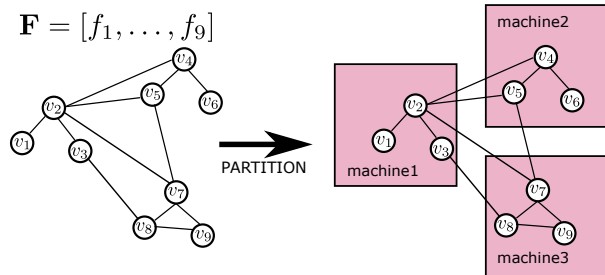

Figure 1: Example of partitioning a graph with 9 nodes into 3 machines. Each machine only stores the features of the nodes in their corresponding partition.

In this paper, we consider the problem of efficiently learning a GNN across a set of machines, each of which has access to part of the graph data (see Figure 1). Drawn from the observation that in a GNN, model parameters are significantly smaller than the graph's input and intermediate node feature, we propose to compress the intermediate GNN node features that are communicated between different machines. Given that this compression affects the accuracy of the GNN, in this paper we introduce a variable compression scheme that trades off the communication overhead needed to train a GNN distributively and the accuracy of the GNN.

The contributions of this paper are:

1. We present a novel algorithm to learn graph representations while compressing the data communicated between the training agents. We propose to vary the compression rate progressively, to achieve a comparable performance to the no-compression case at a fraction of the communication cost.

2. We theoretically show that our method converges to a first-order stationary point of the full graph training problem while taking distributed steps and compressing the inter-server communications.

3. We empirically show that our method attains a comparable performance to the full communication training scenario while incurring fewer communication costs. In particular, by plotting accuracy as a function of the communication costs, our method outperforms full communication and fixed compression rates in terms of accuracy achieved per communicated byte.

## Related work

**Mini-Batch Training.** In the context of distributed GNN training, Zheng et al. (2020) proposes to distribute mini-batches between a set of machines, each of which computes a local gradient, updates the GNN, and communicates it back to the server. Zheng et al. (2020) uses METIS (Karypis and Kumar, 1998) to partition the graph, which reduces communication overhead and balances the computations between machines. Although (Zheng et al., 2020) provides good results in practice, the GNN does not process data on the full graph, only a partition of it, which can yield sub-optimal results compared to processing the full graph. In

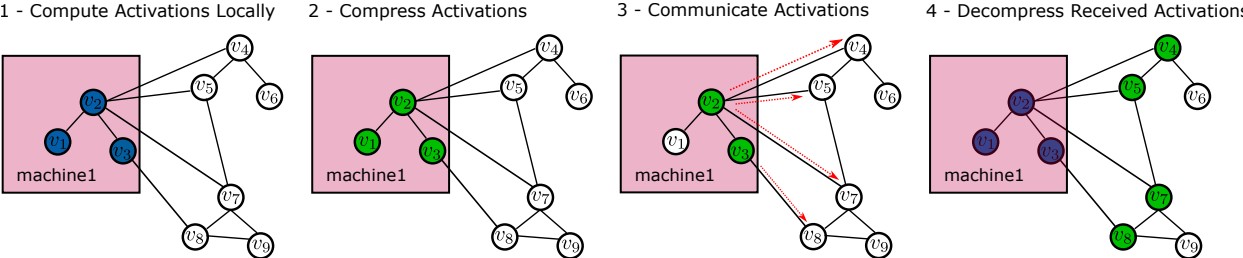

Figure 2: To compute a gradient step, we need to gather the data. To do so, each machine starts by **computing** the activations of the local nodes. Then, these activations are **compressed** and **communicated** to adjacent machines. Once all the activations are communicated, each machine **decompresses** the data from the compressed nodes.

Kaler et al. (2023) they employ a policy to cache data associated with frequently accessed vertices in remote partitions. This method reduces the communications across partitions by creating local copies of the data.

**Memory Optimization.** In this work we do full batch training, which has also been considered in the literature. Similar to us is *sequential aggregation and rematerialization* (Mostafa, 2022), which sequentially re-constructs and frees pieces of the large GNN during the backward pass computation. Even in densely connected graphs, this deals with the memory limitations of the GNN, showing that the memory requirements per worker decrease linearly with the number of workers. Others have studied similar approaches in the context of distributed training (Mostafa et al., 2023). In Md et al. (2021) they propose to use a balanced partitioning of the graph, as well as a shared memory implementation. They utilize a delayed partial aggregation of remote partitions by either ignoring or delaying feature vector aggregation during training.

**Quantization.** A related approach to dealing with the limitations of a single server for training compression in communication is quantization in the architecture. To train GNNs more efficiently using quantization, the most salient examples are feature quantization (Ma et al., 2022), Binary Graph Neural Networks (Bahri et al., 2021), vector quantization (Ding et al., 2021), last bit quantization (Feng et al., 2020), and degree quantization (Tailor et al., 2020). There are also works on adaptive quantization of the messages between machines. In Wan et al. (2023), they adapt the quantization level using stochastic integer quantization. However similar, this work adapts the quantization level locally, at the message level, which differs from our global view of compression. In all, although related in spirit, quantizing a GNN is a local operation that differs from compressing the communication between servers.

**Sampling Based Methods** Our method can be applied in conjunction with sampling-based methods. In sampling-based methods, each node only aggregates from a random subset of its neighbors Zeng et al. (2021); Bai et al. (2021); Serafini and Guan (2021); Liu et al. (2021). In the context of distributed learning, this random subset could include remote nodes. Therefore communication between machines becomes a bottleneck, and our method would still be relevant to reduce the communication volume. If we bias the sampling to only consider local nodes, then this would hurt performance, as it is equivalent to splitting the graph into multiple disconnected components, which does not work well in practice.

## 2 Learning Graph Neural Networks

In this paper, we consider the problem of training GNNs on large-scale graphs that are stored in a set of machines. Formally, a graph is described by a set of vertices and edges $\mathcal{G} = (\mathbf{V}, \mathbf{E})$, where $|\mathbf{V}| = \mathbf{n}$ is the set of vertices whose cardinality is equal to the number of nodes $n$, and $\mathbf{E} \subseteq \mathbf{V} \times \mathbf{V}$ is the set of edges. The graph $\mathcal{G}$ can be represented by its adjacency matrix $\mathbf{S} \in \mathbb{R}^{n \times n}$. Oftentimes, the graph includes vertex features, $F_V \in \mathbb{R}^{|\mathbf{V}| \times |D_v|}$, and edge features, $F_E \in \mathbb{R}^{|\mathbf{E}| \times |D_v|}$, where $D_v$ and $D_E$ are the vertex and edge feature dimensions, respectively. Graph problems fall into three main categories: node-level prediction where the goal is to predict properties of individual nodes; edge-level prediction where the goal is to predict edge features or predict the locations of missing edges; and graph-level prediction where the goal is to predict properties of entire graphs. In this paper, we focus on distributed training of GNNs for node classification

problems. Our distributed training approach is still relevant to the other two types of graph problems, as they also involve a series of GNN layers, followed by readout modules for edge-level or graph-level tasks.

A GNN is a multi-layer network, where at each layer, messages are aggregated between neighboring nodes via graph convolutions. Formally, given a graph signal $\mathbf{x} \in \mathbb{R}^n$, where $[\mathbf{x}]_i$ represents the value of the signal at node $i$, the graph convolution can be expressed as

$$\mathbf{z}_n = \sum_{k=0}^{K-1} h_k \mathbf{S}^k x_n, \tag{1}$$

where $\mathbf{h} = [h_0, \ldots, h_{K-1}] \in \mathbb{R}^K$ are the graph convolutional coefficients. In the case that $K = 2$, and $\mathbf{S}$ is binary, the graph convolution (1) translates into the AGGREGATE function in the so-called *message-passing neural networks*. A GNN is composed of $L$ layers, each of which is composed of a graph convolution (1) followed by a point-wise non-linear activation function $\rho$ such as ReLU, or tanh. The $l$-th layer of a GNN can be expressed as,

$$\mathbf{X}_l = \rho\left( \sum_{k=0}^{K-1} \mathbf{S}^k \mathbf{X}_{l-1} \mathbf{H}_{l,k} \right), \tag{2}$$

where $\mathbf{X}_{l-1} \in \mathbb{R}^{n \times F_{l-1}}$ is the output of the previous layer (with $\mathbf{X}_0$ is equal to the input data $\mathbf{X} = [\mathbf{x}_1, \ldots, \mathbf{x}_n] \in \mathbb{R}^{n \times F_0}$), and $\mathbf{H}_{l,k} \in \mathbb{R}^{F_{l-1} \times F_l}$ are the convolutional coefficients of layer $l$ and hop $k$. We group all learnable coefficients $\mathcal{H} = \{\mathbf{H}_{l,k}\}_{l,k}$, and define the GNN as $\mathbf{\Phi}(\mathbf{x}, \mathbf{S}, \mathcal{H}) = \mathbf{X}_L$.

## 2.1 Empirical Risk Minimization

We consider the problem of learning a GNN that given an input graph signal $\mathbf{x} \subset \mathcal{X} \subseteq \mathbb{R}^n$ can predict an output graph signal $\mathbf{y} \subset \mathcal{Y} \subseteq \mathbb{R}^n$ of an unknown distribution $p(X, Y)$,

$$\underset{\mathcal{H}}{\text{minimize}} \, \mathbb{E}_p[\ell(\mathbf{y}, \mathbf{\Phi}(\mathbf{x}, \mathbf{S}, \mathcal{H}))], \tag{SRM}$$

where $\ell$ is a non-negative loss function $\ell : \mathbb{R}^d \times \mathbb{R}^d \to \mathbb{R}^+$, such that $\ell(\mathbf{y}, \mathbf{y}) = 0$ iif $\mathbf{y} = \mathbf{y}$. Common choices for the loss function are the cross-entropy loss and the mean square error. Problem (SRM) is denoted called Statistical Risk Minimization (Vapnik, 2013), and the choice of GNN for a parameterization is justified by the structure of the data, and the invariances in the graph Bronstein et al. (2017). However, problem (SRM) cannot be solved in practice given that we do not have access to the underlying probability distribution $p$. In practice, we assume to be equipped with a dataset $\mathcal{D} = \{x_i, y_i\}_i$ drawn i.i.d. from the unknown probability $p(X, Y)$ with $|\mathcal{D}| = m$ samples. We can therefore obtain the empirical estimator of (SRM) as,

$$\underset{\mathcal{H}}{\text{minimize}} \, \mathbb{E}_{\mathcal{D}}[\ell(\mathbf{y}, \mathbf{\Phi}(\mathbf{x}, \mathbf{S}, \mathcal{H}))] := \frac{1}{m} \sum_{i=1}^{m} \ell(\mathbf{y}_i, \mathbf{\Phi}(\mathbf{x}_i, \mathbf{S}, \mathcal{H})). \tag{ERM}$$

The empirical risk minimization problem (ERM) can be solved in practice given that it only requires access to a set of samples $\mathcal{D}$. The solution to problem (ERM) will be close to (SRM) given that the samples are i.i.d., and that there is a large number of samples $m$(Vapnik, 2013). To solve problem (ERM), we will resort to gradient descent, and we will update the coefficients $\mathcal{H}$ according to,

$$\mathcal{H}_{t+1} = \mathcal{H}_t - \eta_t \mathbb{E}_{\mathcal{D}}[\nabla_{\mathcal{H}} \ell(\mathbf{y}, f(\mathbf{x}, \mathbf{S}, \mathcal{H}))], \tag{SGD}$$

where $t$ represents the iteration, and $\eta_t$ the step-size. In a centralized manner computing iteration (SGD) presents no major difficulty, and it is the usual choice of algorithm for training a GNN. When the size of the graph becomes large, and the graph data is partitioned across a set of agents, iteration (SGD) requires communication. In this paper, we consider the problem of solving the empirical risk minimization problem (ERM) through gradient descent (SGD) in a decentralized and efficient way.

# 3 Distributed GNN training

Consider a set $\mathcal{Q}, |\mathcal{Q}| = Q$ of workers that jointly learn a single GNN $\boldsymbol{\Phi}$. Each machine $q \in \mathcal{Q}$ is equipped with a subset of the graph $\mathbf{S}$, and node data, as shown in Figure 2. Each machine is responsible for computing the features of the nodes in its local partitions for all layers in the GNN. The GNN model $\mathcal{H}$ is replicated across all machines. To learn a GNN, we update the weights according to gradient descent (SGD), and average them across machines. This procedure is similar to the standard `FedAverage` algorithm used in *Federated Learning*(Li et al., 2020b).

The gradient descent iteration (SGD) cannot be computed without communication given that the data in $(\mathbf{y}, \mathbf{x})_i$ is distributed in the $Q$ machines. To compute the gradient step (SGD), we need the machines to communicate graph data. What we need to communicate is the data of each node in the adjacent machines; transmit the input feature $x_j$ for each neighboring node $j \in \mathcal{N}_i^k, j \in q'$. For each node that we would like to classify, we would require all the graph data belonging to the $k$-hop neighborhood graph. This procedure is costly and grows with the size of the graph, which renders it unimplementable in practice.

As opposed to communicating the node features, and the graph structure for each node in the adjacent machine, we propose to communicate the features and activation of the nodes at the nodes in the boundary. Note that in this procedure the bits communicated between machines do not depend on the number of nodes in the graph and the number of features compressed can be controlled by the width of the architecture used (see Appendix A). The only computation overhead that this method requires is computing the value of the GNN at every layer for a given subset of nodes using local information only.

## 3.1 Computing the Gradient Using Remote Compressed Data

Following the framework of communicating the intermediate activation, to compute a gradient step (SGD), we require 3 communication rounds (i) the forward pass, in which each machine fetches the feature vectors of remote neighbors and propagates their messages to the nodes in the local partition, (ii) the backward pass, in which the gradients flow in the opposite direction and are accumulated in the GNN model weights, and (iii) the aggregation step in which the weight gradients are summed across all machines and used to update the GNN model weights. The communication steps for a single GNN layer are illustrated in Figure 2.

To compute a gradient step, we first compute the forward pass, for which we need the output of the GNN at a node $i$. To compute the output of a GNN at node $i$, we need to evaluate the GNN according to the graph convolution (1). Note that to evaluate (1), we require access to the value of the input features $x_j$ for each $j \in \mathcal{N}_i^k$, which might not be on the same client as $i$. In this paper, we propose that the clients with nodes in $\mathcal{N}_i^k$, compute the forward passes *locally* (i.e. using only the nodes in their client), and communicate the compressed activations for each layer $l$.

Once the values of the activations are received, they are decompressed, and processed by the local machine, and the output of the GNN is obtained. To obtain the gradient of the loss $\ell$, the output of the GNN is compared to the true label, and the gradient with respect to the parameters of the GNN is computed. The errors are once again compressed and communicated back to the clients. Which, will update the values of the parameters of the GNN, after every client updates the values of the parameters $\mathcal{H}$, there is a final round of communication where the values are averaged. Note that this procedure allows the GNN to be updated using the whole graph. The communication costs are reduced given that the number of bits communicated is controlled by the compressing-decompressing mechanism. The compressing and decompressing mechanism can be modeled as follows,

**Definition 1** *The compression and decompression mechanism $g_{\epsilon,r}, g_{\epsilon,r}^{-1}$ with compression error $\epsilon$, and compression rate $r$, satisfies that given a set of parameters $\mathbf{x} \in \mathbb{R}^n$, when compressed and decompressed, the following relation holds i.e.,*

$$\mathbf{z} = g_{\epsilon,r}(\mathbf{x}), \ \text{and} \ \tilde{\mathbf{x}} = g_{\epsilon,r}^{-1}(g_{\epsilon,r}(\mathbf{x})) \ \text{and} \ \mathbb{E}[\|\tilde{\mathbf{x}} - \mathbf{x}\|] \leq \delta \ \text{with} \ \mathbb{E}[\|\tilde{\mathbf{x}} - \mathbf{x}\|^2] \leq \epsilon^2, \tag{3}$$

*where $\mathbf{z} \in \mathbb{R}^{n/r}$ with $n/r \in \mathbb{Z}$ is the compressed signal. If $\delta = 0$, the compression is lossless.*

---

**Algorithm 1** `VARCO`: Distributed Graph Training with **VAR**iable **CO**mmunication Rates

---

Split graph $\mathcal{G}$ into $Q$ partitions and assign them to each worker $q_i$

Initialize the GNN with weights $\mathcal{H}_0$ and distribute it to all workers $q_i$

Fix initial compression rate $c_0$, and scheduler $r$

**repeat**

  **Each Worker** $q_i$: Compute the activations for each node in the local graph (cf. equation (2)) using the local nodes.

  **Each Worker** $q_i$: Compress the activations of the nodes that are in the boundary using the compression mechanism (cf. equation (3)), and communicate them to the adjacent machines.

  **Each Worker** $q_i$: Collect data from all adjacent machines, and compute forward pass by using non-compressed activations for local nodes and compressed activations for non-local nodes that are fetched from other machines.

  **Each Worker** $q_i$: Compute parameter gradients by back-propagating errors across machines and through the differentiable compression routine. Apply the gradient step to the parameters in each worker(cf. equation (SGD)).

  **Server**: Average parameters, send them back to workers, and update compression rate $c_{t+1}$

**until** Convergence

---

Intuitively, the error $\epsilon$, and compression rate $r$ are related; a larger compression rate $r$ will render a larger compression error $\epsilon$. Also, compressed signals require less bandwidth to be communicated. Relying on the compression mechanism (3), a GNN trained using a *fixed* compression ratio $r$ during training, will converge to a neighborhood of the optimal solution. The size of the neighborhood will be given by the variance of the compression error $\epsilon^2$. The first-order compression error $\delta$ is related to the mismatch in the compressed and decompressed signals. Our analysis works for any value of $\delta$, which encompasses both lossy ($\delta > 0$), as well as loss-less compression ($\delta = 0$).

**AS1** *The loss $\ell$ function has $L$ Lipschitz continuous gradients, $||\nabla\ell(\mathbf{y}_1, \mathbf{x}) - \nabla\ell(\mathbf{y}_2, \mathbf{x})|| \leq L||\mathbf{y}_1 - \mathbf{y}_2||$.*

**AS2** *The non-linearity $\rho$ is normalized Lipschitz.*

**AS3** *The GNN, and its gradients are $M$-Lipschitz with respect to the parameters $\mathcal{H}$.*

**AS4** *The graph convolutional filters in every layer of the graph neural network are bounded, i.e.*

$$||h_{*\mathbf{S}}x|| \leq ||x||\lambda_{max}\left(\sum_{t=0}^{T} h_t\mathbf{S}^t\right), \ \ with \ \lambda_{max}\left(\sum_{t=0}^{T} h_t\mathbf{S}^t\right) < \infty. \tag{4}$$

Assumption 1 holds for most losses used in practice over a compact set. Assumption 2 holds for most non-linearities used in practice over normalized signals. Assumption 3 is a standard assumption, and the exact characterization is an active area of research (Fazlyab et al., 2019). Assumption 4 holds in practice when the weights are normalized.

**Proposition 1 (Convergence of GNNs Trained on Fixed Compression)** *Under assumptions 1 through 4, consider the iterates generated by equation (SGD) where the normalized signals $\mathbf{x}$ are compressed using compression rate $c$ with corresponding error $\epsilon$(cf. Definition 1). Consider an $L$ layer GNN with $F$, and $K$ features and coefficients per layer respectively. Let the step-size be $\eta \leq 1/L_\nabla$, with $L_\nabla = 4M\lambda_{max}^L\sqrt{KFL}$ if the compression error is such that at every step $k$, and any $\beta > 0$,*

$$\mathbb{E}_{\mathcal{D}}[||\nabla_{\mathcal{H}}\ell(y, \Phi(x, \mathbf{S}; \mathcal{H}_t))||^2] \geq L_\nabla^2\epsilon^2 + \beta, \tag{5}$$

*then the fixed compression mechanism converges to a neighborhood of the first-order stationary point of SRM in $K \leq \mathcal{O}(\frac{1}{\beta})$ iterations, i.e.,*

$$\mathbb{E}_{\mathcal{D}}[||\nabla_{\mathcal{H}}\ell(y, \Phi(x, \mathbf{S}; \mathcal{H}_t))||^2] \leq L_\nabla^2\epsilon^2 + \beta. \tag{6}$$

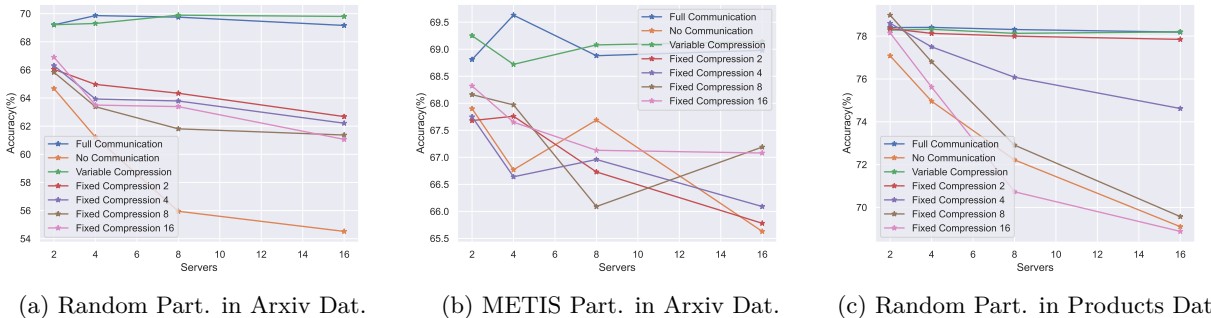

(a) Random Part. in Arxiv Dat.    (b) METIS Part. in Arxiv Dat.    (c) Random Part. in Products Dat.

Figure 3: Accuracy as a function of the number of servers.

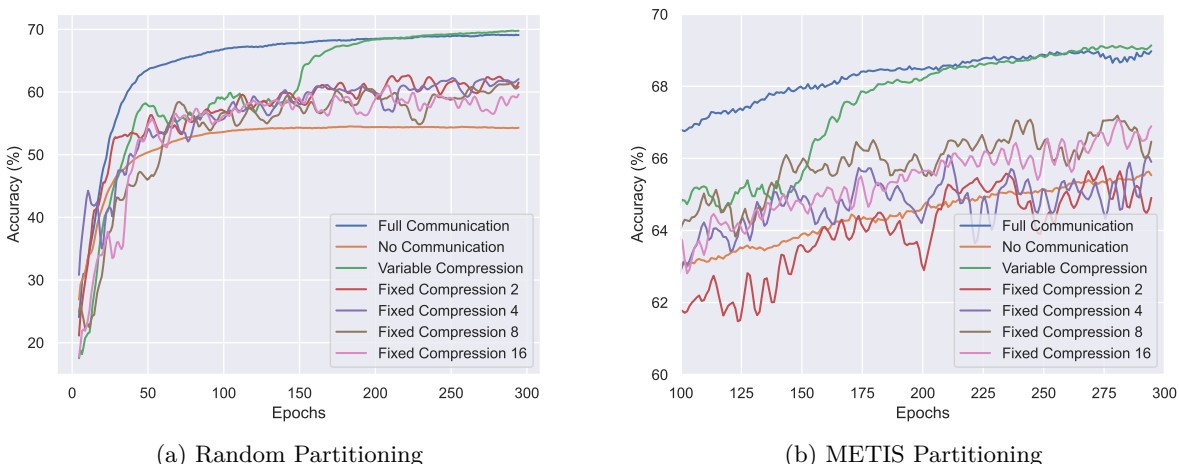

(a) Random Partitioning           (b) METIS Partitioning

Figure 4: Accuracy as a function of epoch for the Arxiv Dataset with 16 servers.

**Proof:** The proof can be found in Appendix E         ■

Proposition 1 is important because it allows us to show that the fixed compression mechanism can converge to a neighborhood of the first-order stationary point of 1. The size of the neighborhood can be controlled by the compression rate $\epsilon$. Although useful, Proposition 1, presents a limitation on the quality of the solution that can be obtained through compression. In what follows we introduce a variable compression scheme that can trade-off between efficient communication and sub-optimality guarantees.

## 4 VARCO - Variable Compression For Distributed GNN Learning

In this paper, we propose variable compression rates as a way to close the gap between training in a centralized manner and efficient training in a distributed manner. We use proposition (1) as a stepping stone towards a training mechanism that reduces the compression ratio $r_t$ as the iterates progress. We begin by defining a scheduler $r(t) : \mathbb{Z} \to \mathbb{R}$ as a strictly decreasing function that given a train step $t \in \mathbb{Z}$, returns a compression ratio $r(t)$, such that $r(t^{'}) < r(t)$ if $t^{'} > t$. The scheduler $r$ will be in charge of reducing the compression ratio as the iterates increase. An example of scheduler can be the linear $r_{lin}(t) = \frac{c_{min} - c_{max}}{T} t + c_{max}$ scheduler (see Appendix 2). In this paper, we propose to train a GNN by following a compression scheme by a scheduler $r$, given a GNN architecture, a number of clients $Q$, and a dataset $\mathcal{D}$.

During the forward pass, we compute the output of the GNN at a node $n_i$ using the local data, and the compressed data from adjacent machines. The compressed data encompasses both the features at the adjacent

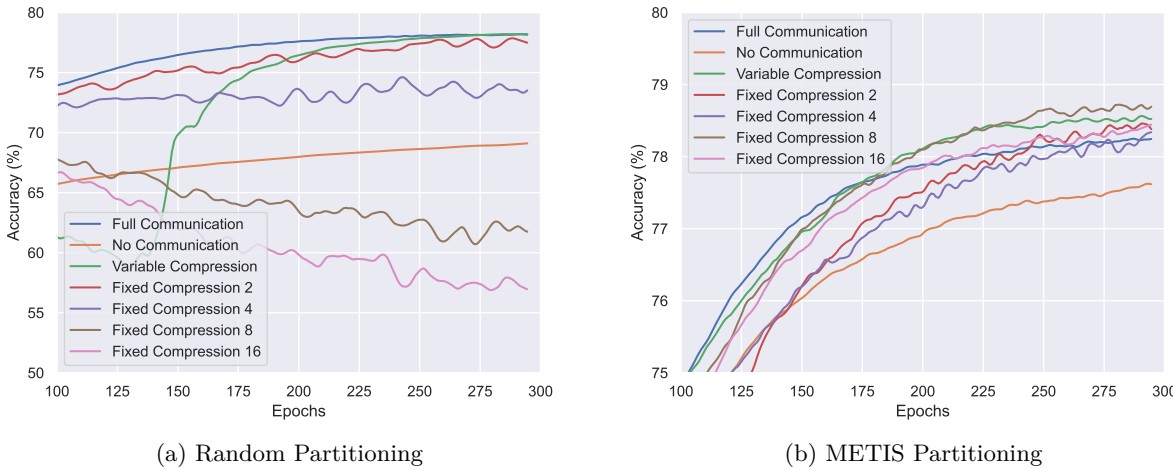

(a) Random Partitioning
(b) METIS Partitioning

Figure 5: Accuracy per epoch for the Products Dataset with 16 servers.

nodes, as well as the compressed activations of the intermediate layers for that node. The compression mechanism compresses the values of the GNN using scheduler $r(t)$, and communicates them the to machine that owns node $n$. The backward pass receives the gradient from the machine that owns node $n$ and updates the values of the GNN. After the GNN values are updated, the coefficients of the GNN are communicated to a centralized agent that averages them and sends them back to the machines. A more succinct explanation of the aforementioned procedure can be seen in Algorithm 1.

## 4.1 Convergence of the VARCO Algorithm

We characterize the convergence of the `VARCO` algorithm in the following proposition.

**Proposition 2 (Scheduler Convergence)** *Under assumptions 1 through 4, consider the iterates generated by equation* (SGD) *where the normalized signals* $\mathbf{x}$ *are compressed using compression rate* $r$ *with corresponding error* $\epsilon$(cf. *Definition 1). Consider an* $L$ *layer GNN with* $F$, *and* $K$ *features and coefficients per layer respectively. Let the step-size be* $\eta \leq 1/L_\nabla$, *with* $L_\nabla = 4M\lambda_{max}^L\sqrt{KFL}$. *Consider a scheduler such that the compression error* $\epsilon_t$ *decreases at every step* $\epsilon_{t+1} < \epsilon_t$, *then for any* $\sigma > 0$

$$\mathbb{E}_{\mathcal{D}}[||\nabla_{\mathcal{H}}\ell(y, \Phi(x, \mathbf{S}; \mathcal{H}_t))||^2] \leq \sigma. \tag{7}$$

*happens infinitely often.*

**Proof:** The proof can be found in Appendix A.1. ∎

According to Proposition 2, for any scheduler, we can obtain an iterate $t$, whose gradient has a norm smaller than $\sigma$. This is an improvement to the fixed compression Proposition 1, given that we removed the term that depends on $\epsilon^2$, converging to a smaller neighborhood. The condition on the scheduler is simple to satisfy; the compression error $\epsilon(t)$ needs to decrease on every step (see more information about schedulers in Appendix A.1). This means that the scheduler does not require information about the gradient of the GNN. Compressing the activations in a GNN can prove to reduce the communication required to train it, given that the overall communication is reduced. However, keeping a fixed compression ratio might not be enough to obtain a GNN of comparable accuracy to the one trained using no compression. By using a variable compression for the communications, we obtain the best of both worlds – we reduce the communication overhead needed to train a GNN, without compromising the overall accuracy of the learned solution. The key observation is that in the early stages of training, an estimator of the gradient with a larger variance is acceptable, while in the later stages, a more precise estimator needs to be used. This behavior can be translated into efficiency; use more information from other servers only when needed.

# 5 Experiments

We benchmarked our method in 2 real-world datasets: OGBN-Arxiv (Wang et al., 2020) and OGBN-Products (Bhatia et al., 2016). In the case of OGBN-Arxiv, the graph has $169,343$ nodes and $1,166,243$ edges and it represents the citation network between computer science arXiv papers. The node features are 128 dimensional embeddings of the title and abstract of each paper Mikolov et al. (2013). The objective is to predict which of the 40 categories the paper belongs to. In the case of OGBN-Products, the graph represents products that were bought together on an online marketplace. There are $2,449,029$ nodes, each of which is a product, and $61,859,140$ edges which represent that the products were bought together. Feature vectors are 100 dimensional and the objective is to classify each node into 47 categories. For each dataset, we partition the graph at random and using METIS partitioning (Karypis and Kumar, 1997) and distribute it over $\{2, 4, 8, 16\}$ machines. In all cases, we trained for 300 epoch. We benchmarked `VARCO` against full communication, no intermediate communication, and fixed compression for $\{2, 4, 8, 16, 32, 64\}$ fixed compression ratios. For the GNNs, we used a 3 layered GNN with 256 hidden units per layer, `ReLU` non-linearity, and `SAGE` convolution (Hamilton et al., 2017). For `VARCO`, we used a linear compression with slopes $\{2, 3, 4, 5, 6, 7\}$, and 128 and 1 maximum and minimum compression ratio respectively (see Appendix A.1). We empirically validate the claims that we put forward, that our method (i) attains the same accuracy as the one trained with full communication, (ii) is more efficient in terms of communication, and (iii) is robust to the choice of the scheduler.

## 5.1 Accuracy

We report the accuracy over the unseen data, frequently denoted test accuracy. In terms of accuracy, we can compare the performance of our variable compression algorithm, against no communication, full communication, and fixed compression ratios. In Figure 3 we show the accuracy as a function of the number of servers for the different baselines considered. We show results for both random 3a and METIS 3b partitioning for the Arxiv dataset, and random partitioning for the products dataset 3c. As can be seen in all three plots, the variable compression attains a comparable performance to the one with full communication. Also, the fixed compression scheme is not able to recover the full communication accuracy when the number of servers increases. Consistent with Proposition 1, as the fixed compression increases, the accuracy attained by the GNN decreases.

We study the accuracy as a function of the number of epochs with 16 servers. This setup is the most challenging, given that the size of the graph in each server is the smallest, and it is therefore the one in which communication is needed. In Figure 4, we show the accuracy per epoch for the Arxiv dataset. As can be seen in both random 4a, and METIS 4b partitioning, the accuracy of variable compression is comparable to the one with full communication. Also, the different fixed compression mechanisms have a worse performance (10% and 3% in random and METIS partitioning respectively), and their performance degrades as their fixed compression increases. In Figure 5, we plot the results for the products dataset with 16 servers. Again, in both partitioning schemes, our variable compression algorithm attains a comparable performance to the one trained on full communication. In this case, compared to the Arxiv dataset 4, the spread of the results is smaller, and the effect of our method is less significant. This is related to the fact that the graph is larger, and therefore, the partitions are also larger. In all, for both partitioning methods, and both datasets, we can validate that the variable compression mechanism attains a comparable performance to the one trained on the full communication, which is not the case for fixed compression.

## 5.2 Efficiency

In terms of efficiency, we can plot the accuracy as a function of the number of floating point numbers communicated between servers. The fixed compression and full communication schemes communicate a constant number of bytes in each round of communication. This number is proportional to the cross-edges between machines, multiplied by the compression coefficient, which in the case of full communication is one. Our method is particularly useful given that at the early stages of training, fewer bytes of communication are needed, and the GNN can be trained with local data only. Intuitively, all learning curves show a similar slope at the beginning of training, and they converge to different values in the later stages. In Figure 6 we

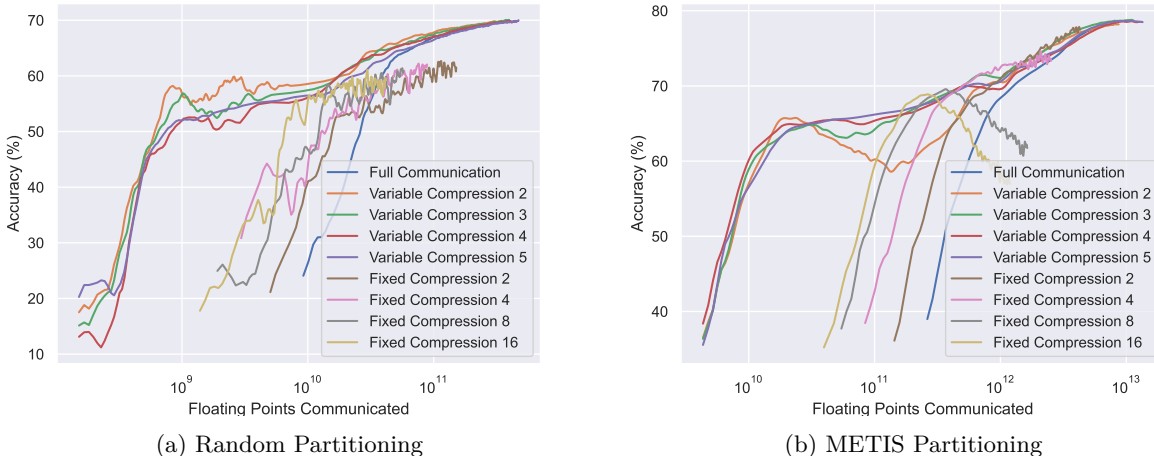

Figure 6: Accuracy per floating points communicated for the products dataset with 16 servers.

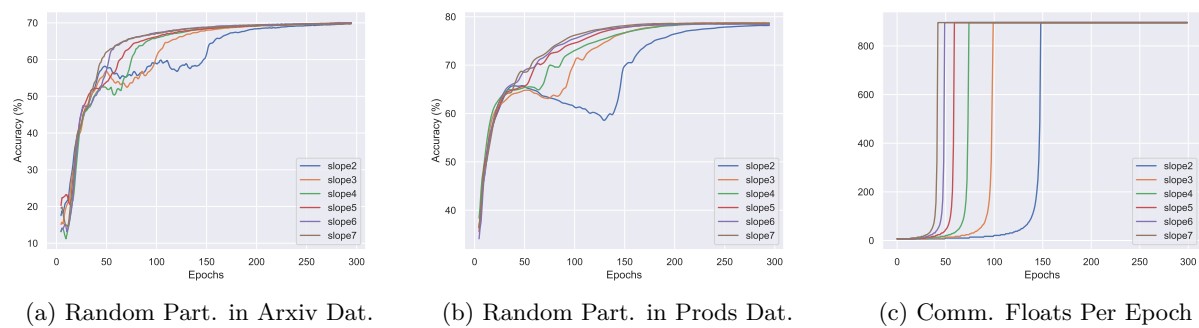

Figure 7: Accuracy per epoch with 16 servers for different variable compression schemes.

corroborate that our method attains the best accuracy as a function of bytes communicated throughout the full trajectory. Given that the VARCO curve in Figure 6 is above all curves, for any communication budget i.e. number of bits, VARCO obtains the best accuracy of all methods considered. This indeed validates our claim that using VARCO is an efficient way of training a GNN.

### 5.3 Robustness

In Figure 7 we validate the robustness of our method to the choice of compression rate. Using linear compression rate, at epoch $e$, with $c_{min} = 1$, $c_{max} = 128$, $E = 300$ and compression rate $c = \min(c_{max} - a\frac{c_{max} - c_{min}}{E}e, c_{min})$, we vary the slope $a$ of the scheduler and verify that the algorithm attains a comparable performance for all runs. This is consistent with Proposition 2, as we only require that the scheduler decreases in every iteration. In Appendix A, the equations that govern the compression rate are described.

## 6 Conclusion

In this paper, we presented a distributed method to train GNNs by compressing the activations and reducing the overall communications. We showed that our method converges to a neighborhood of the optimal solution, while computing gradient estimators communicating fewer bytes. Theoretically, we showed that by increasing the number of bits communicated (i.e. decreasing the compression ratio) as epochs evolve, we can decrease the loss throughout the whole training trajectory. We also showed that our method only requires

the compression ratio to decrease in every epoch, without the need for any information about the process. Empirically, we benchmarked our algorithm in two real-world datasets and showed that our method obtains a GNN that attains a comparable performance to the one trained on full communication, at a fraction of the communication costs.

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
