# OpenReview forum: "Distributed Training of Large Graph Neural Networks with Variable Communication Rates"
_TMLR — Rejected by TMLR_

### Review · Reviewer_huXc · 2024-01-29

**Summary Of Contributions:**

This paper's objective lies in the area of large-scale training of Graph Neural Networks (GNNs), specifically in distributed learning through the utilization of multiple compute nodes (i.e. machines). The authors propose a new approach with the goal of optimization of communication efficiency. The key contribution is the development of a variable compression scheme, called VARCO, which dynamically adjusts compression rates during training to balance communication overhead and model accuracy.

Along with the methodological contribution, the authors present theoretical results of how their method is associated with a full graph training instance. Specifically, they show that the variable compression schema achieves tighter convergence bounds (Equations 7, and 8) than the fixed compression one, which are, also, independent of the compression error $\epsilon(t)$.

Finally, VARCO method is benchmarked in two datasets (a small-size one: ogbn-arxiv #nodes < 200k, # edges $\approx$ 1m, and a mid-size one: ogbn-products: #nodes $\approx$ 2.5m , # edges $\approx$ 60m) against full communication of the compute nodes, no communication, intermediate communication, and fixed compression rates. The authors show that their method is performing similarly to the full communication training, while being more efficient with respect to the communication rate.

**Audience:**

Yes

**Broader Impact Concerns:**

There is no clear concern on the ethical implications of the works that would require a Broader Impact Statement.

**Claims And Evidence:**

Yes

**Requested Changes:**

Questions:
1. To my knowledge, the default versions of ogbn-arxiv, and ogbn-products are directed and undirected graphs respectively. Can the authors clarify whether they used the default versions or not?

In the case of a **directed** graph, the states of a pair of adjacent nodes (with one directed graph) would be not similarly updated, because of the lack of the one edge direction. Would that affect the effectiveness of the third, and fourth steps of Algorithm 1?

2. Figures 4,5,6 show a couple of patterns between the sparser (products) and the denser graph  (arxiv). Specifically, reasonably the no communication case performs better than the higher fixed compression rates (Figure 5), mainly due to the small number of adjacent communication messages required. But, also, it is interesting how the partitioning is affecting the performance of the methods.

It would be very useful to draw an analysis of the convergence of the methods **with respect to the density** of the graphs. Probably a case with degenerate extremely sparse, and extremely dense graphs (even in small size) will be very insightful.

3. The authors are utilizing a specific line of GNNs for their methodology, specifically the convolution-based one.

Can the authors mention if possible how Algorithm 1 could change in the case of more sophisticated architectures, e.g. attention-based GNN? If this is not possible, at least it should be mentioned in the paper, that the methodology should require modifications for training more sophisticated architectures.

4. Although I am not well educated in the area of distributed GNN training, I was able to find a couple of relevant works in the are of communication efficient distributed GNN training. The authors should at least mention them, and indicate their differences:

a. SALIENT++ paper: https://proceedings.mlsys.org/paper_files/paper/2023/file/74e22712c9b50a9b43b2ae54e225888e-Paper-mlsys2023.pdf.

b. ADAQP paper: https://i.cs.hku.hk/~cwu/papers/brwan-mlsys23.pdf

c. DistGNN paper: https://aiichironakano.github.io/cs653/Vasimuddin-DistGNN-SC21.pdf

**Strengths And Weaknesses:**

***Strengths***

1. Interesting approach on variable communication rates, that iteratively updates compression rates of the workers.
2. Solid theoretical results, indicating the contribution of a variable communication schema. Specifically, the key contribution makes the convergence bound independent of the compression error.
3. Empirical success in maintaining accuracy compared to the full communication of learning workers. Also, the method is able to reduce communication costs by attaining strong accuracies with less floating points communicated.
4. The paper is clear and well-written.

***Weaknesses***

See the ***Requested Changes*** review section for an elaborate list of my questions and suggestions. Below, I present a brief list of the same points:
1. It is unclear how the graph characteristics affect the variable compression efficiency. The paper benchmarks two datasets with varied density, for which the presented method, and the fixed compression one perform differently. It would be very insightful to have a better view on how VARCO converges with respect to the density, and the size of the graph.
2. Only one variant of GNNs is studied. It would be interesting to showcase any generalization capabilities of VARCO for diverse GNN types.
3. Related strong baselines are missing. See the next section for more information.

---

> ### Author Response · Authors · 2024-02-20
> **Response**
>
> We thank the Reviewer for their detailed and valuable feedback on our work. We answered each of the Reviewer's questions and explained how we addressed them in the revised version of our manuscript.
>
> ## Answer To Question 1
> The default version of the ogbn-arxiv dataset is directed. During pre-processing, we add the reverse edges to make the graph bi-directional. The default version of ogbn-products is undirected so there was no need to add reverse edges (as every edge has a corresponding reverse edge going in the opposite direction). Making the graph bidirected boosts the performance in ogbn-arxiv as it allows information to flow in both directions between a pair of connected nodes. Making the graph bidirectional thus allows Steps 3 and 4 to propagate data in both directions between a pair of connected nodes. Steps 3 and 4 strictly follow the topology of the graph when propagating forward activations and backward gradients. We thank the reviewer for this question.
>
> ## Answer To Question 2
> The reviewer's question is regarding the relationship between the density of the graph, and the partition method. When the graph is more sparse, and there are fewer edges between machines, the convergence is faster than when the graph is more dense, and there are more edges between machines. The reviewer's point is very interesting, and it is related to the number of cross-edges between machines. When the percentage of cross-edges is large, the role of compression is more significant given that it helps close the gap between full communication and no communication algorithms. As the number of cross-edges decreases, the accuracy of the GNN improves, and the role of compression is less significant. We have added a section in the Appendix addressing this subject. We thank the reviewer for suggesting this analysis. We also note that METIS partitioning, which minimizes the number of edges between the partitions, reduces the advantage of compression since only a few messages (along the inter-partition edges) need to be communicated between the machines, so dropping these messages in the no-communication case does not strongly affect the performance compared to the random partitioning case. In random partitioning, there are a lot of inter-partition edges, and the messages that propagate along these edges thus play a more significant role.
>
> ## Answer To Question 3
> This is a good point. For our experiments, we have used convolutional layers because the theoretical analysis is based on them. However, the algorithm also works for attention layers, with no modification needed. Graph Attention Networks (GATs) also work by having each node aggregate information from its neighboring nodes. The only difference from the networks we use in the paper is that in a  GAT layer, each node uses a separate,  learnable, attention coefficient to weigh the contributions of each of its neighbors before summing these contributions. We will mention in the paper that our approach is also applicable to GAT.
>
> ## Answer To Question 4
> The reviewer mentions three relevant papers. We have added a comparison to them in the body of the paper. We thank the reviewer for pointing this out.

---

> > ### Author Response · Authors · 2024-04-11
> > **Following-Up**
> >
> > We kindly ask the reviewer if any of their concerns remain.

---

### Review · Reviewer_NBMu · 2024-02-06

**Summary Of Contributions:**

This paper presents a distributed GNN training algorithm based on graph partitioning. The proposed method optimizes the training cost by compressing the neighbor information (intermediate activation) to be exchanged among servers in each GNN layer. Theoretical analysis bounds the GNN convergence quality by the compression error. Motivated by the intuition that accurate neighbor aggregation is more desirable when model converges, a variable compression scheme is proposed that gradually decreases the compression ratio when training progresses. Experiments on two standard benchmarks are conducted to validate the accuracy and efficiency of the proposed approach.

**Audience:**

Yes

**Claims And Evidence:**

No

**Requested Changes:**

Please clarify the points in weakness.

In addition, we can make background a bit more concise (e.g., empirical risk minimization is a bit out of topic).

**Strengths And Weaknesses:**

## Strengths

+ The paper is in general well written, with clear motivation & sufficient background.
+ The experiments are evaluated under a number of different settings & metrics (e.g., different number of machines & training epochs, etc).
+ Theoretical analysis has been performed to deepen the technical strength.

## Weaknesses

- The compression & decompression algorithm used in this paper are not clearly defined. From my understanding, compression might mean aggregating local neighbors without visiting remote ones. If this is the case, how would decompression work? In addition, for an MPNN, each layer only aggregates 1-hop neighbors. Then the node should send its own activation to remote neighbors. Why an additional local aggregation is needed?
- The connection between theoretical results and the model design is weak. The 1st theoretical result show that model converges to a better solution when the compression error is smaller. This conclusion is intuitive, but it does not help us design the compression algorithm to achieve such better convergence quality. The 2nd theoretical result show that a schedule with monotonically decreasing compression rate is better. It seems to hold for any monotonically decreasing scheme, and does not justify the specific one used in experiments.
- The experiments are only conducted on relatively small graphs. A single GPU can easily hold the full graph of ogbn-arxiv and ogbn-products. This experimental setting differs from the realistic scenario that motivate the paper.
- The experimental metric is unclear. e.g., what does "accuracy" mean for all the convergence curves? Training accuracy? Test accuracy?
- In the motivation, it should be noted that aggregating from all neighbors may not necessarily produce the best result. Some works (e.g., [1]) deliberately aggregate from a subset of most important neighbors to achieve higher accuracy than full neighbor aggregation.

-----

[1] Decoupling the depth and scope of graph neural networks. In NeurIPS 2021.

---

> ### Author Response · Authors · 2024-02-20
> **Reponse**
>
> We thank the Reviewer for their detailed and valuable feedback on our work. Below, we discuss each of the weaknesses pointed out by the Reviewer. We also explain how we addressed them in the revised version of our manuscript.
>
> ## Reply to Weakness 1
> The Reviewer raises a good point regarding the compression mechanism. By compression, we mean compressing along the feature dimension. For example, a node might have 4 neighbors, 2 residing in the local partition/machine, and 2 remote. For this node to aggregate (take the average of) its 4 neighbors, we compress the features of the 2 remote nodes (for example, from a feature dimension of 256 down to a dimension of 16), send the compressed features, then decompress the features of these 2 remote nodes back to the original feature dimension of 256. We then take the average of the 4 nodes: 2 of them  (the local nodes) will have exact features and 2 will have approximate features (because they have undergone compression and decompression). This is done at every layer.
>
> ## Reply to Weakness 2
> The Reviewer questions the connection between the theoretical results and the model design. First, the idea that a model converges to a better solution when the compression error is smaller is intuitive, and we agree. But, our method uses this intuitive idea as a stepping stone. To exploit this idea, intuitively we need to train the GNN at the highest possible compression rate that still allows the training loss to decrease. Once the GNN gets 'stuck' at a certain training loss, we decrease the compression. In other words, we try to always train at the highest possible compression until convergence. This is the link between Proposition 1, Proposition 2, and efficiency.
>
> The idea that the method works for any compression scheme is precisely what we show in robustness section 5.3. We show that our method works for a wide variety of schedulers, and as the reviewer points out, not just one. We believe this is a benefit of our method, that it only requires a monotonically decreasing scheme. We do not claim that we picked the best scheduler, we pick a family of them, and the results hold for all of them. We, therefore, empirically show the claims that we put forward.
>
> ## Reply to Weakness 3
> We agree with the Reviewer that these datasets are not the largest ones. However, they allow us to create a simple and intuitive example that illustrates the merits of our proposed method.
>
> ## Reply to Weakness 4
> We meant test accuracy. We have added this detail to the body of the paper. We thank the reviewer for pointing this out.
>
> ## Reply to Weakness 5
> We agree with the Reviewer that the mentioned work is a relevant one. Our method would still be applicable there. In sampling-based methods, each node only aggregates from a random subset of its neighbors. This random subset could very well include remote nodes, and our method would still be relevant there to reduce the communication volume. If we bias the sampling to only consider local nodes, then this would hurt performance, as it is equivalent to splitting the graph into multiple disconnected components, which does not work well in practice. Hence, communication is necessary. We have added this explanation to the body of the paper. We thank the reviewer for pointing this out.

---

> > ### Author Response · Authors · 2024-04-11
> > **Following-Up**
> >
> > We kindly ask the reviewer if any of their concerns remain.

---

### Review · Reviewer_itT6 · 2024-02-06

**Summary Of Contributions:**

This work is motivated to boost the performance of distributed-training GNNs to be close to full-communication GNNs, via proposing varying communication rates between servers (or compression error). It provides experimental results of the proposed method, together with theoretical arguments about a gap between convergence of fixed and varying communication rates.

**Audience:**

Yes

**Broader Impact Concerns:**

No additional concern is needed.

**Claims And Evidence:**

Yes

**Requested Changes:**

Please see the above concerns, for both experiments and theory.

**Strengths And Weaknesses:**

Pros:

1. The motivation and methodology look good.

1. In Figure 4 and 5, the experimental benefit of the proposed method is clear from all baselines. (See Concern 2 below for more details.)

Concerns:

1. As pointed out in Section 5.1, the benefit of the proposed method is less clear on larger graphs (OGBN-Products) than on smaller graphs (OGBN-Arxiv). Does this phenomena question the motivation of this work? Or does this imply more servers are necessary for larger graphs?

1. For Figure 4 and 5, some curves are not converging to stable values at epoch 300. Although it might be expensive, it would be good to to train all variants to stationary points for fair comparison.

1. In definition 1, it includes a first-order compression error $\delta$, but it seems not to appear in any following discussion outside of the definition. It would be good to clarify the term more clearly.

1.  Is there any insight about how to ensure that the key assumption in Prop 1 hold? If I understand correctly, the first two terms on the RHS seem to be constant, which means the guarantee of getting close to a stationary point is in order $O(1)$.

1. The proof of Prop 1 in Appendix D may need further improvement:
    1. In Lemma 1 and Lemma 2, the inputs are low-case $x_1,x_2$. Are these standing for a input for single nodes or multiple nodes? From  Eq(2), GNN's input is defined as features of all nodes in the computation tree, as a upper-case $X$. So lower-case $x$'s are not correct as the input of a GNN, as the $L$-layer GNN computes on multiple nodes.

    1. In Eq(15), the gradient of $\rho(x)$ is assumed to be 1 for any $x$?

    1. From Eq(18,19) to (20), a term of $\\|x_1\\|,\\|x_2\\|$ is missing.

    1. In Eq(27,28), which order of norms of these tensors are used?

    1. From Eq(45) to (46), it is not guaranteed that $\\|a+b\\|^2 \ge $ or $\le \\|a\\|^2 + \\|b\\|^2$, which depends on the angle between $a$ and $b$.

    1. In Eq(47), the last term is not zero. Because $\mathbb{E}[a] = \mathbb{E}[b]$ does not imply $\mathbb{E}[a^2] = \mathbb{E}[b^2]$.

1. In Prop 2, is the upper bound still a constant? It will be good to clarify how this can be changed to a small value.

1. (very minor) The figures shall be modified in colors and line styles that are more friendly to readers.

Typos:
1. Under Eq(2), it should be $X=[x_0,x_1,\dots,x_n]^{\top}\in\mathbb{R}^{n \times F_0}$.
1. In Eq(15), $\nabla_{k_v}$ should be $\nabla_{h_v}$.
1. In Eq(24,25), subscripts of $H_1, H_2$ are missing.
1. IN Prop 1 and 2, $\epsilon$ shall be referred to Definition 1 instead of 3.

---

> ### Author Response · Authors · 2024-02-20
> **Reponse**
>
> We thank the Reviewer for their detailed and valuable feedback on our work. Below, we discuss each of the concerns pointed out by the Reviewer. We also explain how we addressed them in the revised version of our manuscript.
>
> ## Reply to Concern 1
> The Reviewer's question is regarding the effect of the size of the graph on our method. As pointed out in section 5.1, the size of each partitioned graph does affect the accuracy of the GNN.
>
> The local graphs are sampled from the whole graph. If the local partitions are larger, they better resemble the large graph. This is related to the transferability properties of GNNs discussed in the paper [Ruiz 2020, Maskey 2023, 2022].
> That is, as the local graphs become larger, the communication with other machines has a smaller impact on the performance of the learned GNN. This is why, if there are fewer number of partitions,  communication is less important.
>
> Returning to the Reviewer's question, the transferability properties do not question the motivation of this work for two reasons. First, even if the number of partitions is small, there will still be a degradation in performance. This degradation can be solved using our method. Second, the number of partitions is not necessarily controlled in practice, as data might be distributed according to geographical constraints. This implies that methods that rely on local data only might not be useful in some situations.
>
> ## Reply to Concern 2
> Thank you for raising this concern. We trained for a number of epochs that we deemed reasonable and practical. We will add additional plots that train for more epochs (until convergence). Once the experiments are ready, we will add them to the paper. Thank you for this comment.
>
> ## Reply to Concern 3
> The Reviewer raises a question regarding the first-order error $\delta$. We have explained it further after Definition 1. In short, the first-order error is related to the expected reconstruction error of the method. Our analysis works for both lossy ($\delta>0$), as well as loss-less ($\delta=0$) compression. We thank the Reviewer for pointing this out.
>
>
> ## Reply to Concern 4
> The Reviewer raises a valid concern regarding Proposition 1. Theoretically, the condition in Proposition 1 is difficult to verify given that we do not have access to the underlying probability distributions. Empirically, this condition assumes that the loss is decreasing in every epoch, which can be measured by taking the loss over data not used for training. This is what motivates decreasing the compression rate.
>
> ## Concern 5
> ### Reply to Concern 5.1
> We corrected this typo. We changed the equations to upper case. Thank you.
> ### Reply to Concern 5.2
> Yes, that is what we meant by normalized Lipschitz in Assumption 3.
> ### Reply to Concern 5.3
> Note that in Proposition 1 we state that signals are normalized. This term that the Reviewer correctly points out is one. We have added the explanation to this step. Thank you.
> ### Reply to Concern 5.4
> In this case, we consider the flattening of the tensor into a vector and therefore define the norm induced by the inner product.
>
> ### Reply to Concern 5.5 and 5.6
> The Reviewer is right in their point. We have modified the proof to simplify the analysis. We have now removed the assumption on the variance of the estimator, and only considered the optimization problem over the dataset.
>
> ## Reply to Concern 6
> The Reviewer's question is regarding the fact that in Proposition 2 we also attain an upper bound on the norm of the gradient squared. We have modified this proof and removed this dependency. We have now considered the case in which we minimize the expected value over the samples, and therefore, the value of $\sigma$ can be as small as needed. We thank the Reviewer for pointing this out, as we believe this modification has improved the overall quality of the paper.
>
> ## Typos
> We thank the Reviewer for pointing out these typos. We have fixed them.

---

> > ### Author Response · Authors · 2024-04-11
> > **Following-Up**
> >
> > We kindly ask the reviewer if any of their concerns remain.

---

### Author Response · Authors · 2024-03-25
**Thank you for the reviews**

We would like to thank the reviewers for their valuable and insightful comments on our work. We are happy with the overall positive feedback regarding the clarity of the presentation, novel contribution, and theoretical results. We hope our responses and changes above address all the main comments that were raised.

We would be happy to engage in further discussions if some queries remain.

---

### Decision · Action_Editor_BCcB · 2024-04-12

**Recommendation:** Reject

**Comment:**

I’m recommending a rejection to encourage the authors to keep improving their method and really demonstrate that the gains in efficiency are worth the cost, e.g. additional complexity in the system, unintuitive training curves, additional things to tune, etc..

The description of the compression algorithm in Appendix A first paragraph should really be moved to the main paper - without this it is hard for a reader to have a concrete understanding about what the compression algorithm is doing.

The compression algorithm might also be improved - instead of filling in 0s for the entries not communicated, maybe some kind of moving average can be kept on each machine and used instead of 0s for those missing features. This might improve your training stability a bit.

**Audience:**

People training large graph neural networks might be interested to read this paper. But the weak empirical evidence might hurt people’s interest a bit.

**Claims And Evidence:**

This paper proposes a new way of training large graph neural networks in a distributed setting based on partitioning the graph and compressing the messages that need to be communicated across the partitions. The authors proposed a variable compression scheme and provided theoretical justifications for it, showing that with a given compression ratio, optimization converges to a neighborhood of the first-order stationary point of training on the full graph.

The claims made in the paper are:
The new training algorithm based on compression, and a variable compression rate scheme that varies the compression rate during training to improve training performance.
Theoretical justification of the algorithm, convergence guarantees.
Empirical evidence from experiment results showing the proposed method can match full graph training with less communication cost.

The evidence provided in the paper mostly supports the claims, but evidence for claim 3 is a bit weak. As a few reviewers pointed out, the experiments were done on not so large graphs, which might actually fit all in one machine. For a simple algorithm like this I think it is needed to have strong empirical evidence, and in particular evidence from cases where such a distributed training algorithm really helps.

The communication cost savings shown in the experiments are also not very significant. Figure 7(c) shows that most of the variable compression rate schedules go to 0 compression very early on during training. Even the method with the least amount of communication cost does half the training with 0 compression. This means at most a 2x saving in communication cost, in which case the training curve actually looks a bit problematic Fig 7(b). The benefit of the proposed algorithm therefore seems relatively small, which comes with its own additional complexities, and overall the evidence is not strong enough that this is actually worth the effort.

**Resubmission Of Major Revision:**

The authors may consider submitting a major revision at a later time.